# tpHusion: An efficient tool for clonal pH determination in *Drosophila*

**Avantika Gupta , Hugo Stocker ***

Institute of Molecular Systems Biology, ETH Zürich, Zürich, Switzerland

\* stocker@imsb.biol.ethz.ch

## Abstract

Genetically encoded pH indicators (GEpHI) have emerged as important tools for investigating intracellular pH (pHi) dynamics in *Drosophila*. However, most of the indicators are based on the Gal4/UAS binary expression system. Here, we report the generation of a ubiquitously-expressed GEpHI. The fusion protein of super ecliptic pHluorin and FusionRed was cloned under the tubulin promoter (tpHusion) to drive it independently of the Gal4/UAS system. The function of tpHusion was validated in various tissues from different developmental stages of *Drosophila*. Differences in pHi were also indicated correctly in fixed tissues. Finally, we describe the use of tpHusion for comparative analysis of pHi in manipulated clones and the surrounding cells in epithelial tissues. Our findings establish tpHusion as a robust tool for studying pHi in *Drosophila*.

## Introduction

Perturbations in intracellular pH (pHi) affect many cellular processes including cell growth, proliferation, differentiation, and metabolism [1]. Under normal conditions, pHi is maintained within a narrow physiological range, and deregulation of pHi is observed in various disease states. In contrast to regular physiology, a state of higher pHi than the extracellular pH (pHe) is conducive to the survival of tumor cells and metastatic progression [2]. Alterations in cytosolic pH have also been implicated to cause neurodegenerative diseases [3]. Exploration of pH dynamics can provide a better understanding of the fundamental cellular processes and help develop strategies for the prevention or treatment of human pathologies.

Over the last few decades, several techniques for pHi measurement have been established. These include the use of proton-permeable microelectrodes, NMR and fluorescence spectroscopy [4,5]. The pH-sensitive fluorescent dyes offer easy-to-use tools for pHi measurement. Although the generation of ratiometric pH probes could address certain shortcomings such as dye leakage, photobleaching and uneven loading of dyes, the application of these dyes is limited in specific cases including pHi analysis in whole-mount tissues because of heterogeneous dye uptake. Construction of the pH-sensitive GFP variant, pHluorin [6], has led to the development of genetically encoded pH indicators (GEpHIs) for pHi measurements in various cell types and organelles. In *Drosophila*, GEpHIs are typically based on the Gal4/UAS system-driven expression of the pH-sensitive supereclipsic pHluorin (SEpHluorin) [7] and the pH-insensitive mCherry [8] in the region of interest [9–11]. Improved modifications of the sensors have also been described [12] but an organism-wide expression has not been exploited. This

**Data Availability Statement:** All relevant data are within the manuscript.

**Funding:** This work was supported by grants from the Swiss National Science Foundation (SNF 31003A_166680; http://www.snf.ch) and the Swiss Cancer League (KLS-3407-02-2014; https://www.

krebsliga.ch) to HS. The funders had no role in study design, data collection and analysis, decision to publish, or preparation of the manuscript.

**Competing interests:** The authors have declared that no competing interests exist.

restricts the ability to examine pHi during developmental processes or in clonal situations within the same tissue, which is one of the major benefits of using *Drosophila* as a model system.

Overexpression of the *Drosophila* Na$^+$-H$^+$ exchanger, Nhe2, increases pHi in the retina [13]. However, the absence of an internal GEpHI expression and imaging control necessitates the generation of fluorescence ratio to pH calibration curves for every experiment. Ubiquitous expression of the GEpHI with a ubiquitously-expressed Gal4 in combination with other means to produce clonal manipulations (FLP/FRT, LexA or Q system) could provide 'in-tissue' control for a more accurate interpretation of results. However, this approach can be complicated by the prevalent use of the Gal4/UAS system and its possible toxicity [14]. To address these issues, we developed a Gal4/UAS-independent and ubiquitously-expressed GEpHI, called tpHusion. We describe its application for evaluation of pHi dynamics in living and fixed tissues, as well as for clonal analysis.

# Materials and methods

## Generation of *tpHusion Act>CD2>Gal4* chromosome

The *tubulin-3'-trailer* was excised from the pKB342 plasmid (gift from Konrad Basler) using restriction sites XhoI and XbaI, and subcloned in-line with the *pHusion-HRas* sequence (gift from Gregory Macleod, [15]) in pJFRC14 vector [16] using BamHI and XbaI sites. The resulting *pHusion-HRas-tubulin-3'-trailer* sequence was excised by restriction digestion with NotI and XbaI and used to replace the *Gal4* sequence under the *tubulin* promoter sequence in the pT2-attB-Gal4 vector (gift from Konrad Basler) using Acc65I and XbaI sites. The resulting plasmid tpHusion was injected into the fly line ΦX-86Fb [17]. The *3xP3-RFPattP* landing site sequence was removed using Cre recombinase-mediated excision. The *Act>CD2>Gal4* construct was recombined on the resulting 3R chromosome arm to generate the *tpHusion Act>CD2>Gal4* chromosome.

## Fly husbandry

All lines and crosses were maintained at 25˚C on normal fly food unless otherwise stated. Normal fly food is composed of 100 g fresh yeast, 55 g cornmeal, 10 g wheat flour, 75 g sugar, 8 g bacto agar, and 1.5% antimicrobial agents (33 g/L nipagin and 66 g/L nipasol in ethanol) in 1 L water.

Fly lines used: *φX-86Fb* (gift from Johannes Bischof), *Act>CD2>Gal4* (4780 Bloomington Drosophila Stock Center (BDSC)); *UAS-ECFP-golgi* (42710 BDSC), *UAS-LacZ* (control, gift from Johannes Bischof), *UAS-Ras$^{V12}$* [18], *Pten$^{Ri}$* (101475 Vienna Drosophila Resource Center (VDRC)), *foxo$^{Ri}$* (107786 VDRC), and *Tsc1$^{Ri}$* (31039 BDSC).

## Cross setup and clone induction

Flies were crossed for two days before an overnight egg laying. For clone induction, a 15 min heat shock at 37˚C was applied 36 h after egg laying (AEL) and the animals were allowed to develop at 25˚C. Wing and eye imaginal discs, and brains were dissected from wandering L3 larvae 108 h AEL; ovarioles were dissected from fertilized females. For clonal analyses, only female larvae were used for dissection. Tissues were dissected in HCO$_3^-$ buffer [19] for live imaging or in 4% paraformaldehyde (PFA) for fixed tissue imaging.

## Microscopy and immunofluorescence staining

After dissection of live tissues in $HCO_3^-$ buffer, the samples were mounted in the same buffer on 35 mm MatTek dishes coated with 0.1 mg/mL poly-L-Lysine. Fluorescence images were acquired on a Visitron Spinning Disk confocal microscope within one hour of dissection.

For fixed samples, tissues were dissected in 4% PFA, fixed for at least 30 min at room temperature (RT), and stored at 4˚C until processing of all experimental conditions. The samples were washed in PBS for 10 min and mounted on glass slides in VECTASHIELD (Vector Laboratories H-1000) mounting medium. Confocal images were obtained within 24 h of dissection on a Leica SPE TCS confocal laser-scanning microscope.

For the immunofluorescence staining with Fasciclin III (FasIII), tissues were dissected in PBS. The samples were fixed in 4% PFA (30 min, RT), washed thrice in 0.3% Triton-X in PBS (PBT, 15 min, RT), blocked in 2% Normal Donkey Serum (NDS) in 0.3% PBT (2 h, 4˚C), incubated with mouse anti-FasIII (1:15 in 2% NDS, 7G10 Developmental Studies Hybridoma Bank (DSHB), overnight, 4˚C), washed thrice in 0.3% PBT (15 min each, RT), incubated with goat anti-mouse Alexa Fluor 647 (1:500, Thermo Fisher Scientific, 2 h, RT), washed thrice in 0.3% PBT (15 min each, RT), stained with DAPI in 0.3% PBT (1:2000, 10 min, RT), and washed once with PBS (10 min, RT). The samples were mounted on glass slides in VECTASHIELD and imaged using a Leica SPE TCS confocal laser-scanning microscope.

## *In vivo* nigericin calibration

Nigericin calibration buffers and curves were generated as described previously [19]. Briefly, after acquiring images of live tissues, the $HCO_3^-$ buffer was replaced with the first calibration buffer containing nigericin. Samples were imaged every 4 min after a minimum incubation of 10 min. After a total incubation time of 20 min, subsequent buffers were added for 6 min and images were acquired every 2 min.

## Quantification and statistical analysis

Images were processed using ImageJ [20]. Background subtraction was performed for each channel. The images were converted to 32-bit and median filtering was applied with radius 2. A defined area encompassing cells of interest, clones or surrounding wild-type tissue was selected and mean gray values were measured for SEpHluorin and FusionRed. The pseudo color images were produced by auto-thresholding of individual channels, followed by division of SEpHluorin channel intensity with FusionRed. The calibration bar represents the relative ratio of SEpHluorin to FusionRed intensities within a tissue from low (blue) to high (red). Statistical analyses were performed using unpaired two-tailed Student's t-test. *p* values are described in the Figure legends. All plots were generated in R Studio and Figures were assembled using Adobe Illustrator.

## Results

### Development and *in vivo* validation of tpHusion pH reporter

The use of the Gal4/UAS system is one of the most common ways to produce clonal manipulations in *Drosophila* [21]. To develop a ubiquitously-expressed GEpHI that would be useful for clonal analysis, the expression of the sensor was rendered independent of Gal4/UAS control. A translational fusion protein of SEpHluorin with FusionRed was cloned under the control of the *tubulin* promoter. FusionRed was used to replace mCherry due to its low cytotoxicity, better performance in fusions and increased stability in a monomeric state [22]. An *HRas* sequence was also included to tether the fusion protein to the cytosolic side of the plasma

membrane. This reduces the pHi variability due to the presence of compartments of differing pH within the cytosol [23]. The construct is referred to as tpHusion. The expression and membrane localization of tpHusion was confirmed in various tissues (data shown for wing imaginal discs in Fig 1A).

To verify that tpHusion is a suitable pH indicator, fluorescence intensity calibrations were performed on wing imaginal discs incubated in buffers containing nigericin, which is an ionophore used to clamp the pHi to the buffer pH (see Materials and Methods). The pH-sensitive SEpHluorin displayed the predicted changes in fluorescence based on the buffer pH, whereas the pH-insensitive FusionRed only showed minor alterations. The ratio of SEpHluorin to FusionRed reflected the buffer pH (Fig 1B), resulting in the generation of a calibration curve of pHi with the ratio of fluorescence intensities (Fig 1C). Thus, tpHusion can be used to indicate changes in pHi.

## tpHusion reports pHi changes in living tissues

Apart from the clonal analyses, a major advantage of a ubiquitously-expressed GEpHI is the ability to compare pHi in different cells across various developmental processes and stages. Earlier studies have reported a higher pHi in the differentiated follicle cells of the *Drosophila* ovariole as compared to the follicle stem cells (FSCs) using the Gal4/UAS-dependent expression of SEpHluorin/mCherry probe [24]. This finding was confirmed using tpHusion, which showed a similar increase in pHi of follicle cells in contrast to the FSCs (Fig 2A and 2A').

The *Drosophila* imaginal discs have proven to be excellent systems for the identification of genes regulating cellular growth during normal development [25,26] or in perturbed states [27,28]. The pHi in these tissues has not been analyzed due to the unavailability of robust indicators. During the larval stages, the eye imaginal disc consists of proliferating cells anterior to the morphogenetic furrow (amf) and mostly differentiating photoreceptors posterior to the furrow (pmf) [29]. Investigation of SEpHluorin and FusionRed intensities using tpHusion demonstrated a higher ratio in the mitotically active amf region of the eye disc (Fig 2B and 2B'). The several compartments and cell lineages of the wing disc have also been described in great detail [26]. The larval wing disc is comprised of cells in the pouch region (which will form the wing blade) and the notum (which will form the body wall) [30,31]. pHi analysis using tpHusion displayed no difference in the pouch versus notum of the wing imaginal disc (Fig 2C and 2C').

A major organ that is vital in neurobiological research is the *Drosophila* brain [32]. Tremendous advances have been made towards deciphering the circuitries of the adult and larval brains [33,34]. The information about pHi of different cell types could be fundamental to understand processes such as vesicular transport [35]. The larval brain can be subdivided into the central brain (CB), optic lobe (OL) and the ventral nerve cord (VNC) [36]. A brief evaluation in the larval brain revealed that cells in the OL have a higher pHi than cells in the CB (Fig 2D and 2D'). The above results establish the use of tpHusion for comparative analysis of *in vivo* pH differences in various *Drosophila* tissues during different developmental stages.

## tpHusion reflects *in vivo* pH changes in fixed tissues

Culturing of *Drosophila* organs has been challenging, with specific culture condition requirements for different tissues [37–40]. For pHi measurements in live cells, tissues are dissected in a bicarbonate buffer (see Materials and Methods) to prevent changes in the physiological pHi. However, the tissues cannot be maintained in this buffer for an extended period, rendering the handling of many experimental conditions difficult. Fixing the conformational state of the GEpHI can help slow down changes in fluorescence until all samples are processed [41]. To

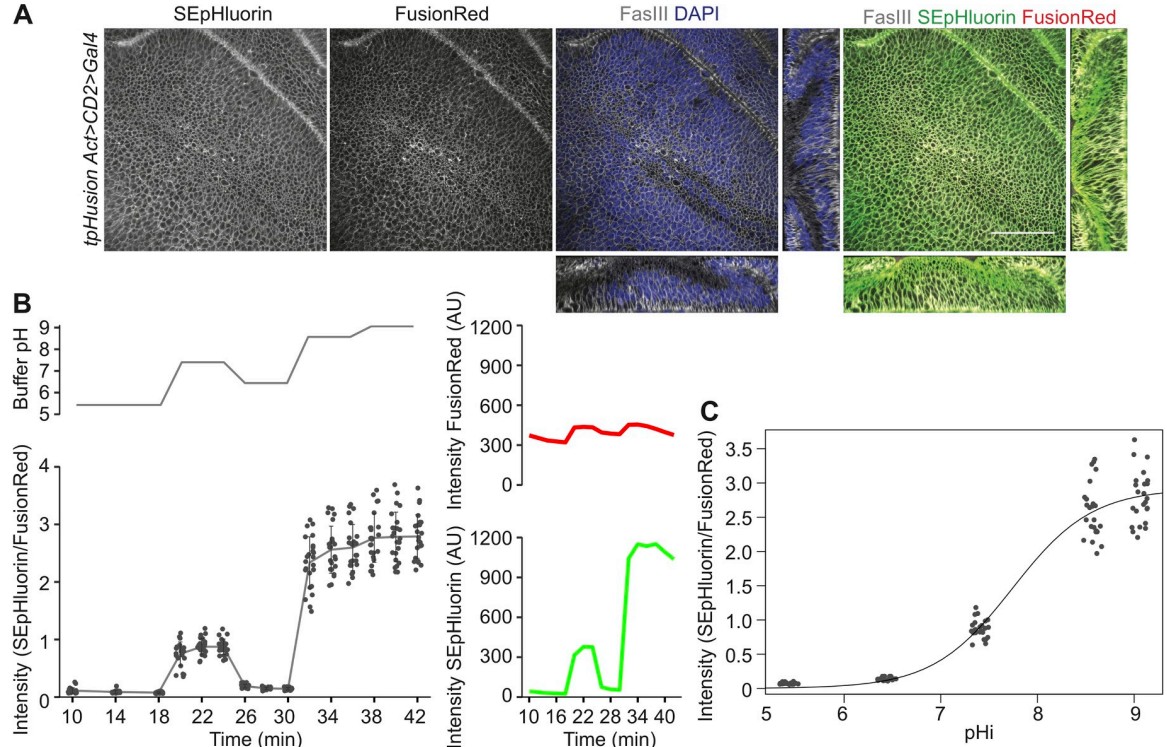

**Fig 1. Validation of tpHusion reporter.** (A) Fixed wing pouches of wandering L3 larvae depicting the cellular localization of tpHusion. FasIII labels the plasma membrane and DAPI stains the nuclei. Scale bar = 50 μm. (B) Changes in SEpHluorin (green) and FusionRed (red) fluorescence intensities, and the ratio of SEpHluorin to FusionRed intensities from live wing pouch cells upon incubation in nigericin buffers of varying pH. n > 7 larvae. Data are represented as mean ± standard deviation. (C) Calibration curve between intracellular pH (pHi) and the ratio of SEpHluorin to FusionRed intensities generated from experiments in B.

examine if tpHusion can be used to monitor pH variations in fixed tissues, comparisons were performed in tissue regions depicted in Fig 2 after fixation. The tissues were dissected directly in 4% PFA to restrict changes in pHi and imaged within 24 h. Interestingly, differences in the ratio of SEpHluorin and FusionRed intensities in the various cell types of the tissues tested were the same as in the live tissues (Fig 3A–3D'). SEpHluorin retains sensitivity to acidic pH after fixation [42]. Since the samples were exclusively exposed to pH 7–7.4 in our experimental setup, the physiological pHi should be maintained. This does not exclude the possibility of slight alterations in pHi but the consistent changes in the ratio of intensities between different regions of the live and fixed tissues (compare Figs 2 and 3) suggest that tpHusion can be reliably used to study pHi variations in fixed tissues.

## Using tpHusion to detect clonal pH changes

Deregulated pH is now considered a hallmark of cancer with a higher pHi and a lower pHe observed in cancer cells as compared to normal cells [43]. Many tumor models have been described in *Drosophila* [44]. Overexpression of activated Ras ($Ras^{V12}$) causes hyperplastic overgrowth [45] and metastatic behavior in combination with loss of polarity genes [27]. It has also been shown to have an increased pHi compared to control cells in a 2D cell culture system of breast epithelial cells [13]. To analyze the pHi in $Ras^{V12}$-overexpressing clones in *Drosophila* epithelial cells, tpHusion was recombined with an actin-FLP-out cassette [46]. Comparison of SEpHluorin and FusionRed intensities ratio in clones and the surrounding wild-type tissue revealed an elevated pHi in the clones (Fig 4A and 4A').

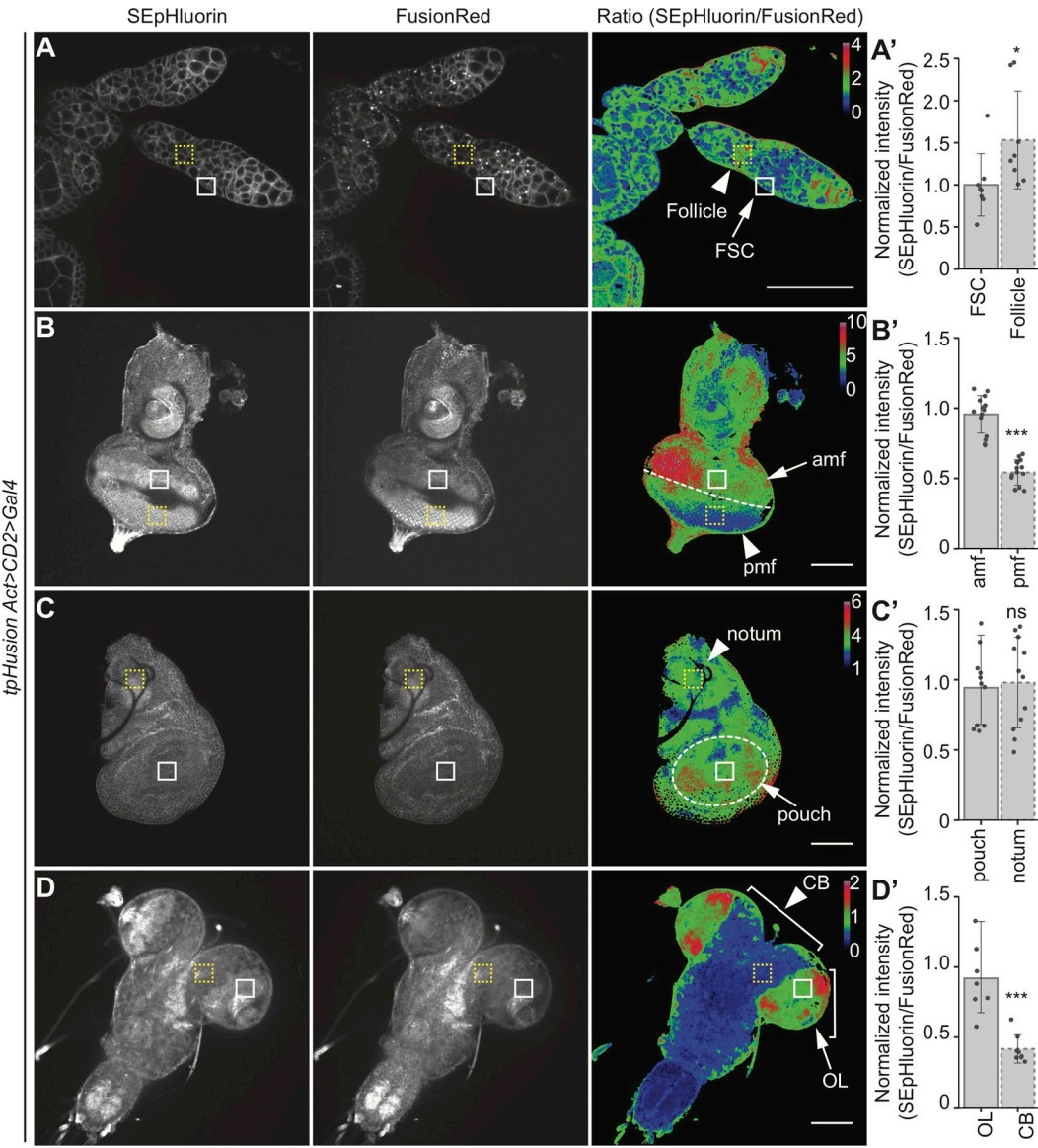

**Fig 2. Comparison of pHi in different cell types of living tissues.** (A-D) SEpHlurin, FusionRed, and ratiometric images of live (A) ovarioles, (B) eye discs, (C) wing discs, and (D) brains. (A'-D') Quantification of the ratio of fluorescence intensities in (A') FSC (arrow, white solid square) and follicle (arrowhead, yellow dashed square) cells, (B') cells anterior (amf, arrow, white solid square) and posterior (pmf, arrowhead, yellow dashed square) to the morphogenetic furrow (dashed line), (C') pouch (arrow, white solid square) and notum (arrowhead, yellow dashed square) cells, and (D') cells in optic lobe (OL, arrow, white solid square) and central brain (CB, arrowhead, yellow dashed square). n > 7 larvae. Data are represented as mean ± standard deviation. * $p < 0.05$, *** $p < 0.001$ and ns = not significant. Scale bar for A = 50 μm, scale bar for B-D = 100 μm. Calibration bar represents the ratio values of SEpHlurin to FusionRed intensities used to generate ratiometric images.

Earlier studies have reported an enhanced cellular overgrowth upon clonal loss of tumor suppressors of the phosphatidylinositol 3-kinase (PI3K)/Akt/mechanistic target of rapamycin (mTORC1) signaling network. Loss-of-function mutations in the phosphatase and tensin homolog (Pten) or the tuberous sclerosis complex (TSC) subunit 1 (Tsc1) lead to an escalation in cell size and number [47,48]. We have also outlined the function of the transcription factor forkhead box O (FoxO) in limiting the proliferation of *Tsc1* mutant cells under conditions of nutrient restriction [49]. The loss of *foxo* does not cause an overgrowth phenotype on its own

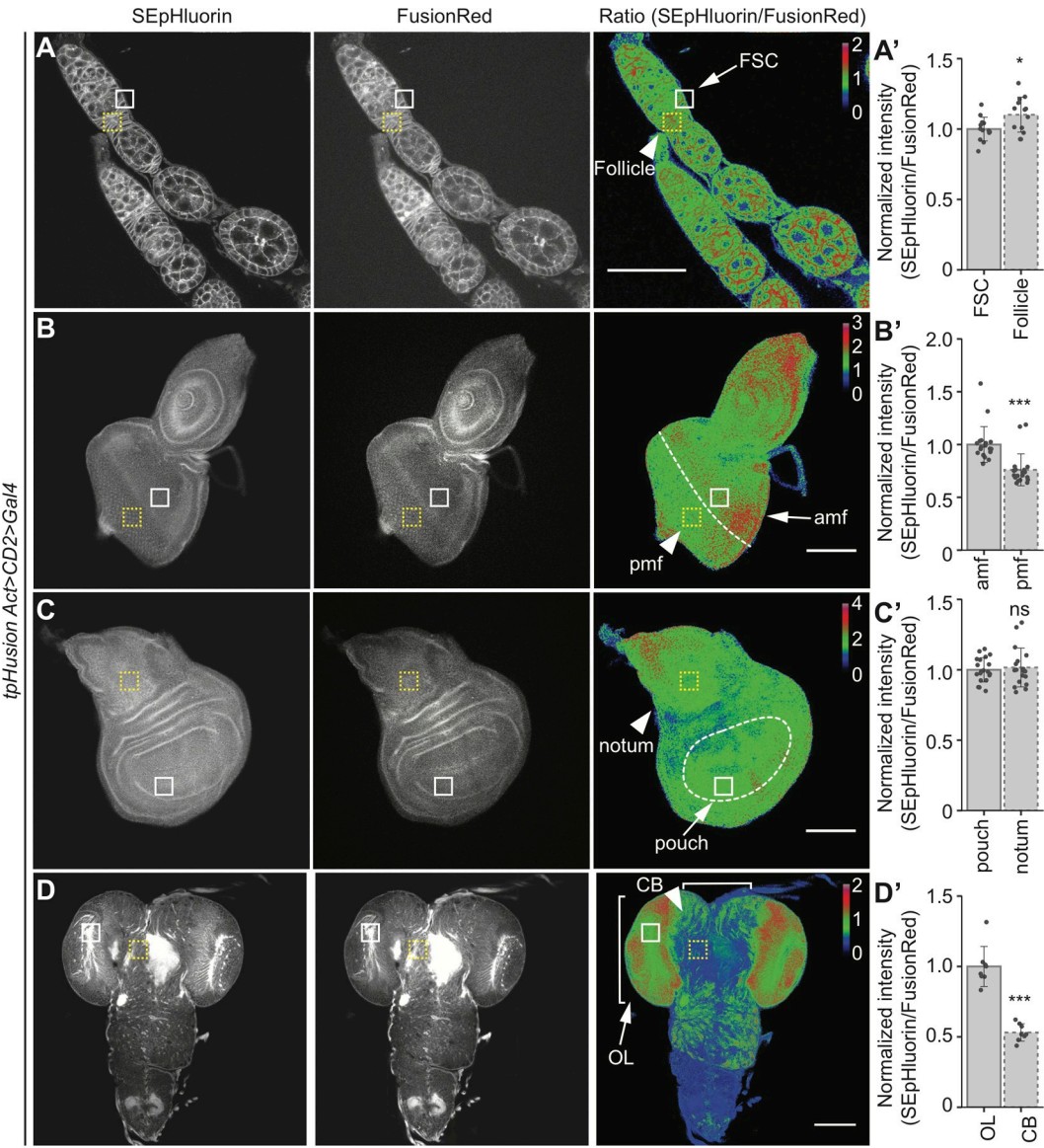

**Fig 3. Comparison of pHi in different cell types of fixed tissues.** (A-D) SEpHluorin, FusionRed, and ratiometric images of fixed (A) ovarioles, (B) eye discs, (C) wing discs, and (D) brains. (A'-D') Quantification of the ratio of fluorescence intensities in (A') FSC (arrow, white solid square) and follicle (arrowhead, yellow dashed square) cells, (B') cells anterior (amf, arrow, white solid square) and posterior (pmf, arrowhead, yellow dashed square) to the morphogenetic furrow (dashed line), (C') pouch (arrow, white solid square) and notum (arrowhead, yellow dashed square) cells, and (D') cells in optic lobe (OL, arrow, white solid square) and central brain (CB, arrowhead, yellow dashed square). n > 7 larvae. Data are represented as mean ± standard deviation. * $p < 0.05$, *** $p < 0.001$ and ns = not significant. Scale bar for A = 50 μm, scale bar for B-D = 100 μm. Calibration bar represents the ratio values of SEpHluorin to FusionRed intensities used to generate ratiometric images.

[50]. Using the system mentioned above, the pHi of control, *Pten*, *foxo*, *Tsc1*, and *Tsc1 foxo* knockdown clones was compared to the surrounding wild-type tissue (Fig 4B). The ratio of SEpHluorin and FusionRed intensities in control and *foxo* knockdown clones was similar to the surrounding wild-type tissue, whereas the ratio was higher in *Pten*, *Tsc1*, and *Tsc1 foxo* knockdown clones. These data validate the use of tpHusion for clonal analysis of pHi in *Drosophila* epithelial tissues and suggest that loss of the tested tumor suppressors increases pHi.

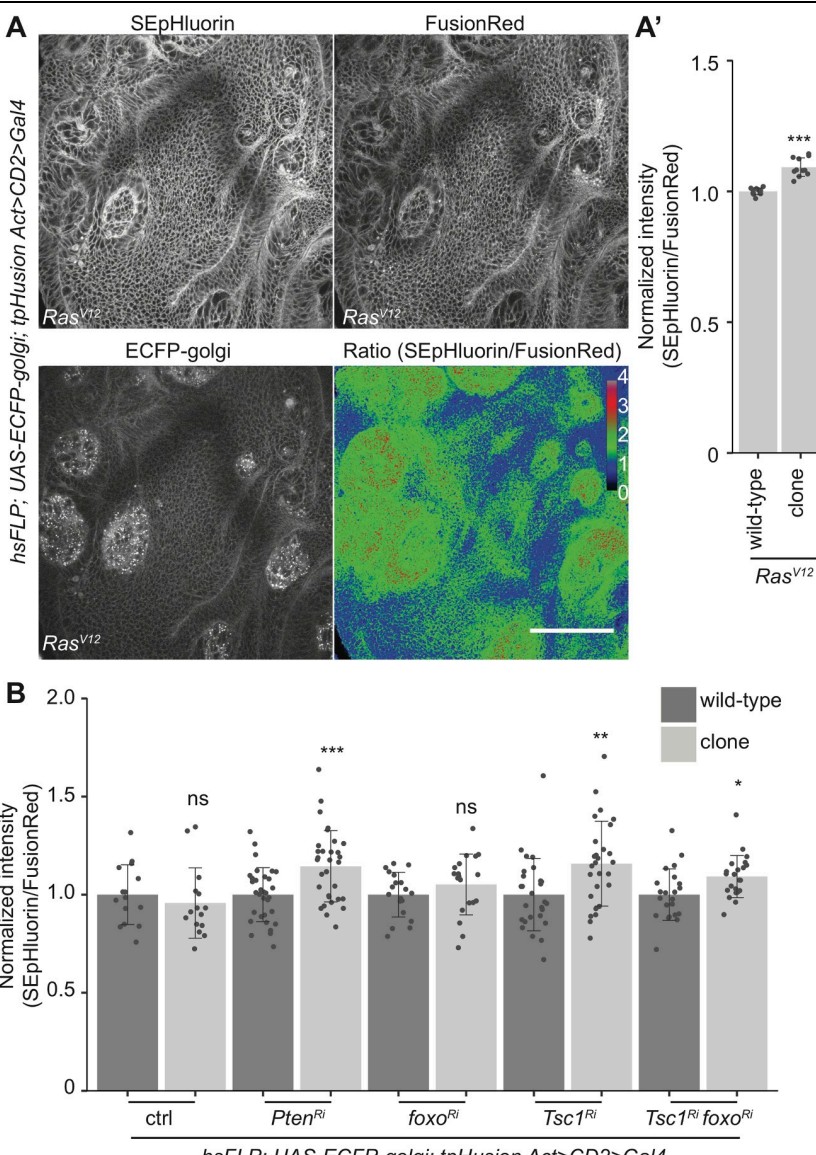

**Fig 4. tpHusion detects clonal pH changes in *Drosophila* imaginal discs.** (A) SEpHluorin, FusionRed and ratiometric images of wing pouches dissected from wandering female L3 larvae with $Ras^{V12}$-overexpressing clones that are marked by ECFP. Calibration bar represents the ratio values of SEpHluorin to FusionRed intensities used to generate ratiometric images. Scale bar = 50 μm. (A') Quantification of the ratio of fluorescence intensities in wild-type and $Ras^{V12}$ clones. (B) Quantification of the ratio of fluorescence intensities in wild-type and control, $Pten^{Ri}$, $foxo^{Ri}$, $Tsc1^{Ri}$ or $Tsc1^{Ri}$ $foxo^{Ri}$ clones from wing pouches of wandering female L3 larvae. n > 10 larvae. Data are represented as mean ± standard deviation. $^{*}$ $p < 0.05$, $^{**}$ $p < 0.01$, $^{***}$ $p < 0.001$ and ns = not significant.

## Discussion

The ubiquitously-expressed GEpHI, tpHusion, facilitates *in vivo* pHi measurement and comparative analysis in diverse tissues and developmental stages in *Drosophila*. We have used this reporter to investigate pHi differences in developing organs and genetically manipulated clones compared to their surrounding cells in the same tissue.

Such an analysis was not feasible with the previously available tools for pHi determination. Our results demonstrate the applicability of tpHusion for the identification of new pHi modulators, and for studying pHi homeostasis during developmental processes or in disease models.

Minor imbalances in the intracellular and extracellular pH equilibrium can have major impacts on many cellular functions. The regulation of cell proliferation by pHi has been shown in a variety of species including sea urchin eggs, yeast, and mammalian cell culture systems [51]. Changing pHi can coordinate differentiation or lineage specification of certain stem cells [24,52,53]. pHi has also been found to change during cell cycle progression [54]. Given the crucial role of pH in physiology, it is not surprising that a loss of pH homeostasis is seen in many diseases including cancer [55]. One notable aspect of this observation that remains disputed in the field is if pH can signal to cellular processes leading to diseases or whether the observed pH imbalance is a result of the pathological state.

Studies addressing the role of pHi in the fundamental processes mentioned above have been limited in *Drosophila* due to the lack of adequate tools. The uniform expression of tpHusion with no apparent cytotoxicity throughout development can aid the analysis of pHi dynamics in various developmental contexts. The complete potential of tpHusion can be realized by combining it with the existing repertoire of genetic modification tools in *Drosophila*. The flexible use of tpHusion with the Gal/UAS and FLP-out systems to express UAS-based transgenes in clones led us to present, for the first time, an increase in pHi in cells with overexpression of an oncogene or knockdown of tumor suppressor genes as compared to the surrounding wild-type cells. One caveat is the inability to use the most commonly used fluorophores to label clones or cell lineages but this can be circumvented by the use of non-interfering fluorophores in the blue or the far-red channels. We conclude that tpHusion is a powerful tool for investigating cellular pH in *Drosophila*.

## Acknowledgments

We are indebted to Michal Stawarski and Gregory Macleod for generously providing the UAS-pHusion plasmid and for valuable comments and inputs. We also thank Bree Grillo-Hill for discussions about imaging in the $HCO_3^-$ buffer and nigericin calibration, and Ryohei Yagi for cloning advice. We are grateful to Johannes Bischof, BDSC and VDRC for flies, Konrad Basler for plasmids, DSHB for the FasIII antibody, Joachim Hehl at ScopeM ETH Zurich for help with live imaging, and Igor Vuillez for technical assistance.

## Author Contributions

**Conceptualization:** Avantika Gupta, Hugo Stocker.

**Formal analysis:** Avantika Gupta.

**Funding acquisition:** Hugo Stocker.

**Investigation:** Avantika Gupta.

**Methodology:** Avantika Gupta.

**Supervision:** Hugo Stocker.

**Writing – original draft:** Avantika Gupta.

**Writing – review & editing:** Hugo Stocker.

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
