## [Decision Letter · Decision Letter 0]

6 Dec 2019

PONE-D-19-31154

tpHusion: An efficient tool for clonal pH determination in Drosophila

PLOS ONE

Dear Hugo,

Thank you for submitting your manuscript to PLOS ONE. Your manuscript has now been evaluated by two expert reviewers. Both reviewers find your work interesting and that it has merit, but also made a few suggestions to further improve it. The editorial decision is "minor revision". Therefore, we invite you to submit a revised version of the manuscript that addresses the points raised during the review process.

We would appreciate receiving your revised manuscript by Jan 20 2020 11:59PM. To enhance the reproducibility of your results, we recommend that if applicable you deposit your laboratory protocols in protocols.io, where a protocol can be assigned its own identifier (DOI) such that it can be cited independently in the future. For instructions see: http://journals.plos.org/plosone/s/submission-guidelines#loc-laboratory-protocols

We look forward to receiving your revised manuscript.

Kind regards,

Andreas

***

Andreas Bergmann, Ph.D.

Academic Editor

PLOS ONE

Journal Requirements:

Additional Editor Comments (if provided):

Reviewers' comments:

Reviewer's Responses to Questions

**Comments to the Author**

1. Is the manuscript technically sound, and do the data support the conclusions?

Reviewer #1: Yes

Reviewer #2: Yes

2. Has the statistical analysis been performed appropriately and rigorously? 

Reviewer #1: I Don't Know

Reviewer #2: Yes

3. Have the authors made all data underlying the findings in their manuscript fully available?

Reviewer #1: Yes

Reviewer #2: Yes

4. Is the manuscript presented in an intelligible fashion and written in standard English?

Reviewer #1: Yes

Reviewer #2: Yes

5. Review Comments to the Author

Reviewer #1: This manuscript by Gupta and Stocker examines the usefulness and efficiency of a new genetic tool that assesses the intracellular pH of cells. The study describes a practical follow-up to pH detectors that are already in use. The major difference of the tool generated in this work is that it functions independent of the Gal4/UAS binary gene expression system. The authors demonstrate the use of tpHusion in live, fixed, and clonal tissues.

Overall, this study is relevant for three reasons. (1) The tpHusion tool can detect intracellular pH in live and fixed tissues independent of Gal4/UAS. (2) tpHusion can be used along with binary expression systems to evaluate intracellular pH in mosaic tissue. (3) tpHusion can be used to assess intracellular pH across different tissues, developmental stages, and biological processes. However, there are a few issues that should be addressed:

(1) The figures and figure legends can be better annotated to highlight different regions of the tissue. Dashed lines, arrowheads, or brackets can be used to indicate various regions within the tissue. For example, on the eye discs, a dashed line along the morphogenetic furrow can more clearly show the amf versus the pmf. Similarly, arrowheads pointing out the FSCs versus the follicle cells allows for better interpretation of the distribution of the tpHusion signal in the ovarioles. The wing disc and the larval brain images can also benefit from outlining and labeling the different compartments.

(2) The dashed squares in the figures are difficult to see. Replace them with solid squares of a different color or represent another way. It would be helpful to have the squares on the images in the single channels too.

(3) Explain the calibration bar a bit more – specifically the colors and the associated numbers.

(4) Increase the magnification of the ovarioles in figure 2. The magnification of the ovarioles differs between figure 2 and figure 3. At the higher magnification used in figure 3, it is easier to see the cells and the tpHusion signal.

(5) Include low magnification images of the single channels of figure 4 for an unbiased view of the distribution of the tpHusion signal across clones throughout the entire wing disc.

(6) The authors state that the detection of intracellular pH in fixed tissue is similar to that observed in live tissue. However, there appears to be a significant reduction of tpHusion signal in the images of the fixed eye discs. Based on the images presented, the tpHusion signal in the fixed larval brain appears expanded and more intense. Finally, the tpHusion signal in fixed ovarioles seems shifted with more intense FusionRed signal compared to SEpHluorin signal in fixed compared to live tissues. Discuss these differences.

Additional minor points

(1) Change HCO3 to HCO3-.

(2) Page 6, Line 122: Insert (NDS) after Normal Donkey Serum.

Reviewer #2: Gupta et al. describe the development of tpHusion, a membrane associated cytosolic pH sensor. The authors describe their strategy, which relies on expressing the sensor under the control of the tubulin promoter in order to obviate the necessity of the GAL4/UAS system. The utility of the strategy was validated by measurements of pH in different tissues during development, and in both fixed as well as live tissues.

The manuscript is technically sound, and the authors’ conclusions are backed by their data. This tool will certainly be useful to researchers interested in studying cytosolic changes during development. There remains, however, one minor concern that could be addressed by additional experiments as detailed below. tpHusion is tagged to the membrane using an HRAS-sequence. Though this is a reasonable strategy to attach the sensor to the PM, there is a concern regarding localization of the sensor to specific domains at the PM. Attachment of HRAS to the membrane is dependent upon the levels of membrane cholesterol (i.e. lipid rafts). Under conditions of alterations in PM cholesterol levels, the sensor could fall off the membrane. How would this change the pH measurements? It is recommended that the authors describe the effects of removing PM cholesterol with beta-cyclodextrin on the recordings made with tpHusion. The goal of this endeavor would be to provide a framework that future researchers could use should they worry about expressing the sensor in backgrounds that exhibit alterations in cholesterol trafficking.

6. PLOS authors have the option to publish the peer review history of their article (what does this mean?). If published, this will include your full peer review and any attached files.

Reviewer #1: No

Reviewer #2: No

---

## [Author Response · Author response to Decision Letter 0]

24 Jan 2020

Please see our Response to Reviewers where we address all concerns in detail (including two figures).

---

## [Editor Report · Decision Letter 1]

29 Jan 2020

tpHusion: an efficient tool for clonal pH determination in Drosophila

PONE-D-19-31154R1

Dear Hugo,

Thank you for submitting your revised manuscript to PLOS ONE. We are pleased to inform you that your manuscript has been judged scientifically suitable for publication and will be formally accepted for publication once it complies with all outstanding technical requirements.

With kind regards,

Andreas

Andreas Bergmann, Ph.D.

Academic Editor

PLOS ONE

---

## [Editor Report · Acceptance letter]

4 Feb 2020

PONE-D-19-31154R1 

tpHusion: an efficient tool for clonal pH determination in *Drosophila*

Dear Dr. Stocker:

I am pleased to inform you that your manuscript has been deemed suitable for publication in PLOS ONE. Congratulations! Your manuscript is now with our production department. 

With kind regards,

on behalf of

Dr. Andreas Bergmann 

Academic Editor

PLOS ONE